# Prediction of Masked Uncontrolled Hypertension Detected by Ambulatory Blood Pressure Monitoring

**DOI:** 10.3390/diagnostics12123156

**Published:** 2022-12-14

**Authors:** Francesca Coccina, Paola Borrelli, Anna M. Pierdomenico, Jacopo Pizzicannella, Maria T. Guagnano, Chiara Cuccurullo, Marta Di Nicola, Giulia Renda, Oriana Trubiani, Francesco Cipollone, Sante D. Pierdomenico

**Affiliations:** 1Department of Innovative Technologies in Medicine & Dentistry, University “Gabriele d’Annunzio”, Chieti-Pescara, 66100 Chieti, Italy; 2Laboratory of Biostatistics, Department of Medical, Oral and Biotechnological Sciences, University “Gabriele d’Annunzio”, Chieti-Pescara, 66100 Chieti, Italy; 3Department of Medicine and Aging Sciences, University “Gabriele d’Annunzio”, Chieti-Pescara, 66100 Chieti, Italy; 4Department of Neurosciences, Imaging and Clinical Sciences, University “Gabriele d’Annunzio”, Chieti-Pescara, 66100 Chieti, Italy

**Keywords:** ambulatory blood pressure, classification, hypertension, masked, prediction

## Abstract

The aim of this study was to provide prediction models for masked uncontrolled hypertension (MUCH) detected by ambulatory blood pressure (BP) monitoring in an Italian population. We studied 738 treated hypertensive patients with normal clinic BPs classified as having controlled hypertension (CH) or MUCH if their daytime BP was < or ≥135/85 mmHg regardless of nighttime BP, respectively, or CH or MUCH if their 24-h BP was < or ≥130/80 mmHg regardless of daytime or nighttime BP, respectively. We detected 215 (29%) and 275 (37%) patients with MUCH using daytime and 24-h BP thresholds, respectively. Multivariate logistic regression analysis showed that males, those with a smoking habit, left ventricular hypertrophy (LVH), and a clinic systolic BP between 130–139 mmHg and/or clinic diastolic BP between 85–89 mmHg were associated with MUCH. The area under the receiver operating characteristic curve showed good accuracy at 0.78 (95% CI 0.75–0.81, *p* < 0.0001) and 0.77 (95% CI 0.73–0.80, *p* < 0.0001) for MUCH defined by daytime and 24 h BP, respectively. Internal validation suggested a good predictive performance of the models. Males, those with a smoking habit, LVH, and high-normal clinic BP are indicators of MUCH and models including these factors provide good diagnostic accuracy in identifying this ambulatory BP phenotype.

## 1. Introduction

Masked uncontrolled hypertension (MUCH), that is, normal clinic but high out-of-office blood pressure (BP), in treated patients has been broadly studied in the past two decades [1,2,3,4,5,6,7,8,9,10,11,12,13,14,15,16,17,18,19,20,21,22,23,24,25,26,27]. This condition may be present in approximately one-third of treated hypertensive patients with normal clinic BPs [1,2,3,4,5,6,7,8,9,10,11,12,13,14,15,16,17,18,19,20,21,22,23,24,25,26,27]. Various underlying potential mechanisms have been described [21,23,28,29,30]. With regards to its clinical impact, single studies and meta-analyses have shown that MUCH has approximately twice the risk of cardiovascular morbidity and mortality compared to controlled hypertension (CH), both in and out-of-office [1,2,3,4,5,6,7,8,9,10,11,12,13,14,15,16,17,18,19,20,21,22,23,24,25,26,27]. This phenomenon can be identified by using either home BP recording [1,2,3,4,5,6,21,22,23,24,25,26,27] or ambulatory BP monitoring [7,8,9,10,11,12,13,14,15,16,17,18,19,20,21,22,23,24,25,26,27]. To define MUCH by ambulatory BP monitoring, preceding studies have applied thresholds of ≥135/85 mm Hg for daytime and/or ≥120/70 mm Hg for nighttime and/or ≥130/80 mm Hg for 24 h BP [7,8,9,10,12,13,14,15,16,17,18,19,20,21,22,23,24,25,26,27]. The global population of treated hypertensive patients with normal clinic BPs should undergo out-of-office BP evaluation to detect MUCH. Though this approach would be desirable, it is not always feasible for various reasons in clinical practice. In this context, it would be useful to find prediction models that could help to select patients eligible for out-of-office BP evaluation. Though factors associated with MUCH have been described in previous studies [1,2,3,4,5,6,7,8,9,10,11,12,13,14,15,16,17,18,19,20,21,22,23,24,25,26,27,31,32,33], few reports [34,35] have attempted to provide prediction models. Thus, other studies could be helpful to add further knowledge on this matter. The aim of this study was to provide prediction models for MUCH detected by ambulatory BP monitoring in an Italian population.

## 2. Materials and Methods

### 2.1. Patients

We studied 738 treated hypertensive patients with normal clinic BPs selected from 2264 sequential treated individuals aged 30 to 90 years who were prospectively recruited from December 1992 to December 2012. All patients had been referred to our hospital outpatient clinic for evaluation of BP control. One hundred and three patients were lost during follow-up. Subjects with secondary hypertension were excluded. All subjects underwent clinical evaluation, electrocardiogram, routine laboratory tests, echocardiographic examination, and non-invasive ambulatory BP monitoring. The study population came from the same geographical area (Chieti and Pescara, Abruzzo, Italy). 

### 2.2. Clinic BP Measurement

Clinic BP was recorded by a physician using a mercury sphygmomanometer and appropriate-sized cuffs. Measurements were performed in triplicate, 2 min apart after at least 5 min of rest, and the mean value was used as the BP for the visit. Clinic systolic BPs (SBP) and diastolic BPs (DBP) were defined as normal when <140/90 mmHg.

### 2.3. Ambulatory BP Monitoring

Ambulatory BP monitoring was performed with a noninvasive recorder (SpaceLabs 90207, Redmond, WA, USA) on a typical day, within 1 week of the clinic visit. Technical aspects have been previously reported [36]. We evaluated the following ambulatory BP parameters: daytime (awake period as reported in the diary), nighttime (asleep period as reported in the diary), and 24 h systolic and diastolic BP. MUCH was defined using the 2018 European Society of Hypertension (ESH) guidelines [37], by a clinic BP of < 140/90 mmHg, and by 2 ambulatory BP definitions: (1) daytime SBP ≥ 135 and/or DBP ≥ 85 mmHg regardless of nighttime BP (that is, nighttime SBP< or ≥ 120 and/or DBP< or ≥ 70 mmHg), (2) 24 h SBP ≥ 130 and/or DBP ≥ 80 mm Hg regardless of daytime or nighttime BP as defined above. All the subjects had recordings of good quality (at least 70% of valid readings during the 24 h period, at least 20 valid readings while awake with at least 2 valid readings per hour and at least 7 valid readings while asleep with at least 1 valid reading per hour), in line with the European Society of Hypertension requirements [38]. 

### 2.4. Echocardiography

Left ventricular (LV) measurements and calculation of LV mass were made according to standardized methods [39]. LV mass was indexed by height^2.7^ and LV hypertrophy was defined as LV mass/height^2.7^ > 50 g/m^2.7^ in men and >47 g/m^2.7^ in women [40]. 

### 2.5. Statistical Analysis

Descriptive analysis was carried out using mean and standard deviation for the quantitative variables and percentage values for the qualitative variables. Normality distribution for quantitative variables was assessed using the Shapiro–Wilk Test. Comparison between CH and MUCH according to various definitions was performed using an unpaired Student’s *t* test for continuous variables and Pearson’s Chi-square or Fisher’s exact test for categorical variables. Univariate and multivariate logistic regression analyses were used to estimate potential predictors of MUCH by odds ratio (OR) and corresponding 95% confidence interval (CI). Variables that were significantly associated with MUCH in univariate analysis were included in multivariate analysis. A prediction scoring system for MUCH was produced using the ORs of predictors. The largest among the ORs of predictors was set as the higher value and the other ORs were rescaled relative to the maximum point. The total score was calculated as the sum of the points assigned to all predictors. A receiver operating characteristic (ROC) curve analysis was used to assess the diagnostic accuracy of the models, including the variables that were associated with MUCH in multivariate analysis and by the created scoring systems. The goodness of fit was evaluated using the Hosmer–Lemeshow test. Sensitivity and specificity were calculated at an optimal cut-off point derived from Youden’s J-index. We also performed internal validation of the models using k-fold cross validation. The original sample was randomly partitioned into 10 equal sized subsamples for ROC curve comparison of the predictive model. The mean ROC for each k-fold was then reported with 95% CI. Statistical significance was defined as *p* < 0.05. Analyses were made with the SPSS 21 (SPSS Inc. Chicago, IL, USA) and STATA 17 (StataCorp, College Station, TX, USA).

## 3. Results

We detected 215 (29%) patients with daytime MUCH regardless of nighttime BP (42 had daytime MUCH only and 173 had daytime and nighttime MUCH) and 275 (37%) patients with 24 h MUCH regardless of daytime or nighttime BP (33 had daytime MUCH only, 69 had nighttime MUCH only and 173 had daytime and nighttime MUCH).

Characteristics of patients with CH, daytime MUCH regardless of nighttime BP, and 24 h MUCH regardless of daytime or nighttime BP are shown in Table 1. Prevalence of men, smokers, and LV hypertrophy was higher in patients with MUCH.

BP values and the therapeutic strategy of study groups are reported in Table 2. Though in the normal range, clinic systolic and diastolic BP and prevalence of high-normal clinic systolic and diastolic BP were higher in patients with MUCH than in those with CH. Daytime, nighttime, and 24 h BPs were higher in patients with MUCH by definition. Single, double, and triple therapy did not differ between the groups.

Use of aspirin and statin was not different between patients with CH and MUCH for each definition (14–16% versus 13–16% and 7–9% versus 6–8%, respectively). In univariate logistic regression analysis, male sex, smoking, LV hypertrophy, clinic systolic BP in the range of 130–139 mmHg, and clinic diastolic BP in the range of 85–89 mmHg were significantly associated with MUCH defined by both thresholds (Table 3).

All variables that were significant in the univariate analysis, except smoking habit for MUCH defined according to 24 h BP threshold which approached significance, remained significantly associated with MUCH (Table 4). The Hosmer–Lemeshow tests suggested a good fit for both models (MUCH defined according to daytime BP, Chi-square = 7.01, *p* = 0.43; MUCH defined according to 24 h BP, Chi-square = 6.0, *p* = 0.65).

Figure 1 shows ROC curves based on the variables included in multivariate analyses in predicting MUCH defined by daytime and 24 h BP thresholds. The CI of the area under the curve (AUC) of the two models was between 0.73 and 0.81.

Figure 2 shows ROC curves based on the score systems in predicting MUCH defined by daytime and 24 h BP thresholds. To predict MUCH defined according to daytime BP threshold, a cutoff of 4.5 points had a sensitivity and specificity of 75% and 68%, respectively. To predict MUCH defined according to the 24 h BP threshold, a cutoff of 3.5 points had a sensitivity and specificity of 71% and 69%, respectively. 

Ten-fold cross validation showed that cross-validated mean AUC, standard deviation, and bootstrap bias corrected 95% CI were 0.77 ± 0.06 and 0.72–0.79, respectively, for the model predicting MUCH defined according to daytime BP threshold, and 0.77 ± 0.05 and 0.73–0.80, respectively, for the model predicting MUCH defined according to 24 h BP threshold (Figure 3).

When we applied to our population the prediction model suggested by Kim et al. [34], the AUC for MUCH defined by 24 h BP threshold was 0.73, 95% CI 0.69–0.77. If we used in our population the best prediction model suggested by Hung et al. [35], although they used a different definition of MUCH and also included patients with masked hypertension, the AUC for MUCH defined by 24 h BP threshold (our definition that most closely matched their definition of MUCH) was 0.72, 95% CI 0.68–0.75.

## 4. Discussion

The present study shows that: (1) male sex, smoking habit, LV hypertrophy, a clinic systolic BP in the range of 130–139 mmHg, and/or a clinic diastolic BP in the range of 85–89 mmHg are associated with MUCH defined by both daytime and 24 h BP thresholds; (2) prediction models based on the abovementioned variables were appropriate in identifying the presence of MUCH; (3) internal validation indicated a good predictive performance of the models. Though characteristics of patients with MUCH have been described in previous studies [1,2,3,4,5,6,7,8,9,10,11,12,13,14,15,16,17,18,19,20,21,22,23,24,25,26,27,31,32,33], few reports [34,35] have attempted to provide prediction models. Kim et al. [34], studied 854 treated hypertensive patients with normal clinic BPs (<140/90 mmHg) enrolled in the Korean Ambulatory BP Monitoring Registry. Among them, 465 had CH and 389 had MUCH defined as 24 h BP < or ≥130 and/or 80 mmHg, respectively. Multivariate logistic regression analysis showed that high clinic systolic and/or diastolic BP, previous stroke, dyslipidemia, LV hypertrophy, high heart rate and single antihypertensive drug use were independent predictors of MUCH. A scoring system based on the strength of association of the abovementioned variables with MUCH in the regression analysis exhibited a good diagnostic accuracy (AUC 0.84, 95% CI 0.81–0.87) for the detection of MUCH. According to their scoring system, a score ≥ 9.6 points had a sensitivity and specificity of 79% and 78%, respectively. Hung et al. [35], evaluated the characteristics of patients that could be able to predict masked hypertension and MUCH. They studied a cohort of 970 hypertensive patients (six medical centers in Taiwan) which were used for model development and internal validation and a cohort of 416 hypertensive patients (one medical center) which was used for external validation. The authors used 33 clinical characteristics as candidate variables to develop models based on logistic regression (LR), random forest (RF), eXtreme Gradient Boosting (XGboost), and artificial neural network (ANN). The four models showed good sensitivity in internal and external validation. The RF (AUC 0.85, 95% CI 0.79–0.91), XGboost (AUC 0.80, 95% CI 0.73–0.87), and ANN (AUC 0.80, 95% CI 0.74–0.87) models showed higher AUC than the LR model (AUC 0.67, 95% CI 0.59–0.76) in internal validation. A similar trend was observed in external validation. However, among the RF, XGboost, and ANN models, the RF including only six predictors (clinic systolic BP, clinic diastolic BP, clinic mean arterial pressure, clinic pulse pressure, beta-blocker use, and high density lipoprotein cholesterol) had the best performance in both internal and external validation. Our study shows similarities and differences in comparison with previous ones [34,35]. Indeed, the AUC of the predicting models of all studies showed good diagnostic accuracy [41] and clinic systolic/diastolic BPs in the high-normal range were included in the models of all the studies, further indicating that these parameters are the most powerful ones. Other variables included in the models were different between our study and previous reports [34,35]. This aspect, however, could be explained by different populations’ characteristics and different ethnicities. Moreover, in the study by Hung et al. [35], masked hypertension and MUCH were analyzed together. When we applied to our population the prediction models suggested by Kim et al. [34] and Hung et al. [35], their diagnostic accuracy tended to be lower. This aspect suggests that our model might be best suited in a Caucasian population, but the other two models might be preferable in Asian populations. When the cut-off values of our scoring system were used to predict MUCH defined by daytime and 24 h BP thresholds, approximately 25% of patients with MUCH were not identified and approximately 30% of those with CH were included, suggesting a reasonable accuracy of the scoring system itself. Only a quarter of patients with MUCH, who given their ambulatory BP and risk should increase treatment (though this strategy is still under investigation in an ongoing trial [42]), were not identified. However, as a systematic search for patients with MUCH is not carried out in many countries, we would like to stress that the proposed approach would at least allow the identification of roughly three quarters of patients at risk, which represents already a big improvement over simply relying on resting office BPs. In this study, we reported a cluster of factors, and their potential weight, that could help to identify patients who are more likely to have MUCH and should undergo out-of-office BP evaluation to unmask this BP phenotype that exposes them to a greater cardiovascular risk and to potentially improve the overall therapeutic strategy. The present study has some limitations. First, we studied only Italian subjects, and our results cannot be directly extrapolated to other ethnic groups. Second, as the prevalence of chronic kidney disease tended to be low and that of diabetes was low in our population, the results cannot be extrapolated to populations of patients with chronic kidney disease [43] or diabetes mellitus [14]. Third, as with any cross-sectional observation, the association between the identified variables and MUCH does not automatically establish a causal relationship. Fourth, it was possible to evaluate adherence to therapy only in a part of the patients. However, a recent study showed that MUCH is not attributable to medication non-adherence [44]. Fifth, we performed an internal validation of the model, but we could not execute an external validation which could have provided more confidence in the model. Our study also has some strengths including (1) a fairly large sample size, (2) consistent findings when comparing the two models and when comparing them with similar studies, and (3) the ability to deliver a relatively easy-to-use model, which could be integrated into routine clinical practice. 

## 5. Conclusions

Male sex, smoking habit, presence of LV hypertrophy, and high-normal clinic systolic and/or diastolic BPs are predictors of MUCH and models including these factors appear to provide good diagnostic accuracy in identifying this ambulatory BP phenotype and show a good predictive performance. The suggested prediction models may be a useful tool to select patients eligible for out-of-office BP evaluation.

## Figures and Tables

**Figure 1 diagnostics-12-03156-f001:**
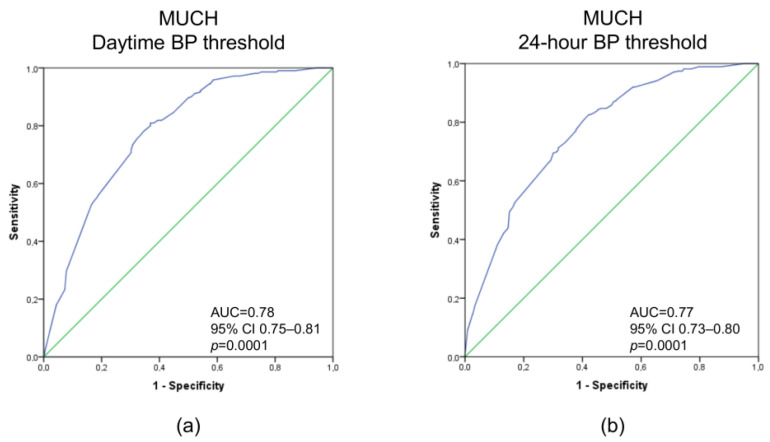
Receiver operating characteristic curves based on variables included in multivariate analyses in predicting masked uncontrolled hypertension defined by daytime blood pressure (BP) threshold (**a**), regardless of nighttime BP, and by 24 h BP threshold (**b**), regardless of daytime or nighttime BP. Abbreviations: AUC, area under the curve; CI, confidence interval.

**Figure 2 diagnostics-12-03156-f002:**
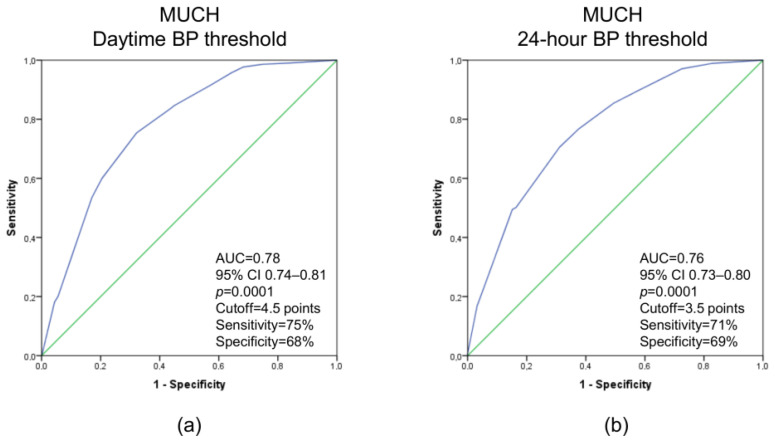
Receiver operating characteristic curves based on the score systems in predicting masked uncontrolled hypertension (MUCH) defined by daytime blood pressure (BP) threshold (**a**), regardless of nighttime BP, and by 24 h BP threshold (**b**), regardless of daytime or nighttime BP. For MUCH defined by daytime BP threshold, the score is as follows: clinic systolic BP 130–139 mmHg = 3 points; clinic diastolic BP 85–89 mmHg = 1.5 points; men = 1 point; smoker = 1 point; left ventricular (LV) hypertrophy = 1 point. For MUCH defined by 24 h BP threshold, the score is as follows: clinic systolic BP 130–139 mmHg = 1.5 points; clinic diastolic BP 85–89 mmHg = 2 points; men = 1 point; smoker = 1 point; LV hypertrophy = 1 point. Other abbreviations: AUC, area under the curve; CI, confidence interval.

**Figure 3 diagnostics-12-03156-f003:**
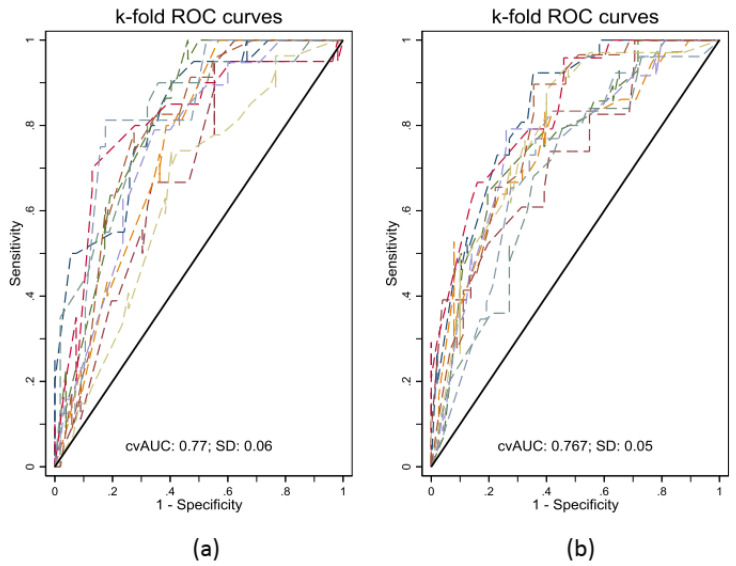
Internal validation of the models using 10-fold cross validation for masked uncontrolled hypertension defined by daytime blood pressure (BP) threshold (**a**), regardless of nighttime BP, and by 24 h BP threshold (**b**), regardless of daytime or nighttime BP. Abbreviations: AUC, area under the curve; SD, standard deviation.

**Table 1 diagnostics-12-03156-t001:** Characteristics of study groups.

Parameter	Daytime BP Threshold		24-h BP Threshold	
	CH	MUCH	CH	MUCH
n.	523	215	463	275
Age, years	61 ± 10	60 ± 11	61 ± 10	60 ± 11
Men, n (%)	202 (39)	126 (59) †	166 (36)	162 (59) †
BMI, kg/m^2^	28 ± 5	28 ± 4.0	28 ± 5	28 ± 4
Smokers, n (%)	84 (16)	54 (25) †	76 (16)	62 (22) *
FHCVD, n (%)	64 (12)	19 (9)	59 (13)	24 (9)
Previous events, n (%)	29 (6)	8 (4)	24 (5)	13 (5)
Diabetes, n (%)	27 (5)	14 (7)	24 (5)	17 (6)
eGFR < 60 mL/min, n (%)	123 (23)	43 (20)	113 (24)	53 (19.5)
LDL cholesterol, mg/dL	129 ± 30	127 ± 28	130 ± 29	126 ± 30
HDL Cholesterol, mg/dL	51 ± 10	50 ± 11	51 ± 10	50 ± 11
Triglycerides, mg/dL	128 ± 56	134 ± 62	128 ± 57	133 ± 61
LV hypertrophy, n (%)	79 (15)	57 (27) †	69 (15)	67 (24) †
Clinic HR, beats/min	74 ± 12	75 ± 12	74 ± 12	76 ± 12

BMI, body mass index; BP, blood pressure; CH, controlled hypertension (below ambulatory BP threshold values, see text); eGFR, estimated glomerular filtration rate; FHCVD, family history of cardiovascular disease; HDL, high density lipoprotein; HR, heart rate; LDL, low density lipoprotein; LV, left ventricular; MUCH, masked uncontrolled hypertension (above ambulatory BP threshold values, see text). * *p* < 0.05, † *p* < 0.01 vs. CH for each classification.

**Table 2 diagnostics-12-03156-t002:** Blood pressure values and therapeutic strategy of study groups.

Parameter	Daytime BP Threshold		24-h BP Threshold	
	CH	MUCH	CH	MUCH
n.	523	215	463	275
Clinic SBP, mmHg	130 ± 7	134 ± 5 †	129 ± 7	133 ± 6 †
Clinic DBP, mmHg	80 ± 6	83 ± 5 †	79 ± 6	83 ± 5 †
Clinic SBP 130–139, n (%)	293 (56)	193 (90) †	253 (55)	233 (85) †
Clinic DBP 85–89, n (%)	221 (42)	157 (73) †	180 (39)	198 (72) †
Daytime SBP, mmHg	121 ± 8	137 ± 7 †	121 ± 8	134 ± 8 †
Daytime DBP, mmHg	75 ± 6	84 ± 7 †	74 ± 6	83 ± 7 †
Nighttime SBP, mmHg	110 ± 11	122 ± 12 †	109 ± 9	123 ± 12 †
Nighttime DBP, mmHg	65 ± 7	72 ± 8 †	63 ± 6	73 ± 7 †
24-h SBP, mmHg	118 ± 8	133 ± 8 †	117 ± 7	131 ± 8 †
24-h DBP, mmHg	72 ± 6	80 ± 7 †	71 ± 6	80 ± 7 †
Single therapy, n (%)	246 (47)	99 (46)	220 (48)	125 (46)
Double therapy, n (%)	194 (37)	79 (37)	167 (36)	106 (38)
Triple therapy, n (%)	83 (16)	37 (17)	76 (16)	44 (16)

BP, blood pressure; CH, controlled hypertension (below ambulatory BP threshold values, see text); DBP, diastolic blood pressure; MUCH, masked uncontrolled hypertension (above ambulatory BP threshold values, see text); SBP, systolic blood pressure. † *p* < 0.01 vs. CH for each classification.

**Table 3 diagnostics-12-03156-t003:** Predictors of MUCH by specific thresholds in univariate analysis.

Parameter	Daytime BP Threshold			24-h BP Threshold		
	OR_C_	95 % (CI)	*p*	OR_C_	95% (CI)	*p*
Men vs. Women	2.25	1.63–3.11	0.0001	2.56	1.89–3.48	0.0001
Smoker vs. non–Smoker	1.75	1.19–2.58	0.004	1.48	1.02–2.16	0.04
LVH vs. no–LVH	2.03	1.38–2.98	0.0001	1.84	1.26–2.68	0.001
CSBP < 120 mmHg	1			1		
CSBP 120–129 mmHg	1.09	0.39–3.10	0.86	0.88	0.41–1.88	0.74
CSBP 130–139 mmHg	7.38	2.90–18.7	0.0001	4.19	2.13–8.23	0.0001
CDBP < 80 mmHg	1			1		
CDBP 80–84 mmHg	0.86	0.49–1.51	0.59	1.14	0.69–1.89	0.61
CDBP 85–89 mmHg	3.44	2.23–5.30	0.0001	4.31	2.86–6.49	0.0001

BP, blood pressure; CDBP, clinic diastolic blood pressure; CI, confidence interval; CSBP, clinic systolic blood pressure; LVH, left ventricular hypertrophy; MUCH, masked uncontrolled hypertension (above ambulatory BP threshold values, see text); OR_C_, crude odds ratio. The other variables reported in Table 1 and the number of antihypertensive drugs were not significantly associated with MUCH.

**Table 4 diagnostics-12-03156-t004:** Predictors of MUCH by specific thresholds in multivariate analysis.

Parameter	Daytime BP Threshold			24-h BP Threshold		
	OR_A_	95 % (CI)	*p*	OR_A_	95% (CI)	*p*
Men vs. Women	1.84	1.28–2.63	0.001	2.20	1.57–3.08	0.0001
Smoker vs. non–Smoker	1.91	1.23–2.96	0.004	1.46	0.96–2.22	0.08
LVH vs. no–LVH	2.01	1.30–3.10	0.002	1.85	1.21–2.83	0.004
CSBP < 120 mmHg	1			1		
CSBP 120–129 mmHg	0.91	0.31–2.65	0.86	0.69	0.31–1.53	0.36
CSBP 130–139 mmHg	5.28	2.02–13.8	0.001	2.72	1.33–5.56	0.006
CDBP < 80 mmHg	1			1		
CDBP 80–84 mmHg	0.62	0.33–1.13	0.12	0.90	0.52–1.54	0.70
CDBP 85–89 mmHg	2.52	1.56–4.07	0.0001	3.42	2.19–5.36	0.0001

BP, blood pressure; CDBP, clinic diastolic blood pressure; CI, confidence interval; CSBP, clinic systolic blood pressure; LVH, left ventricular hypertrophy; MUCH, masked uncontrolled hypertension (above ambulatory BP threshold values, see text); OR_A_, adjusted odds ratio.

## Data Availability

The data underlying this article are available upon reasonable request from the corresponding author.

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
