# Peer review of "Prediction of Masked Uncontrolled Hypertension Detected by Ambulatory Blood Pressure Monitoring"

_diagnostics, 2022, doi:10.3390/diagnostics12123156_

Round 1

Reviewer 1 Report

Francesca Coccina et col.  aimed with this study providing prediction models for masked uncontrolled hypertension (MUCH) detected by ambulatory blood pressure (BP) monitoring in an Italian population. Methods: 738 treated hypertensive subjects with standard clinic BP classified as having controlled hypertension (CH) or MU (masked) CH if daytime BP < or >135/85 mmHg regardless of nighttime BP, respectively, or CH or MUCH if 24-hour BP < or >130/80 mmHg irrespective of daytime or nighttime BP.  The authors conclude that gender (male), smoking, LVH and high-normal clinic BP could be capable of predicting MUCH in this sample. They also provide models, including factors for diagnosing with reasonable accuracy in identifying this ambulatory BP phenotype (Masked Hypertension). Statistics and English are satisfactory. 

This manuscript is fair, the methods well described, and the results somewhat intriguing. The results and conclusion are very well sustained in the Introduction and Objectives. I have a couple of suggestions to improve this current version: 1- The medical importance of the MUCH has not been well exploited  (Introduction and Discussion sections); 2- As a consequence (or cause), the text shows VERY few recent references on the matter. The issue has been "hot" in the last few years and deserves more attention in the paper, such as numbers/percentages in mortalith and morbidity (and other consequences), as well as patophysioly of the MUCH. These topics should be approached in 1-2 additional paragraphs. 3- If MUCH is "a clinical disease" or "a face of clinical disease" (Hypertension), are there advantages "to treat" this aspect of Hypertension? Although parts of items 2 and 3 have been briefly described, these topics should be better clarified in the form of a discussion.  

I thank the authors for sharing their good manuscript with this referee.

Author Response

We thank the Reviewer very much for the kind and constructive revision.

Q: 1-The medical importance of the MUCH has not been well exploited  (Introduction and Discussion sections);

A: We have now tried to better explain the medical importance of MUCH in the Introduction (lines 37-42) and Discussion section (lines 296-299).

Q: 2- As a consequence (or cause), the text shows VERY few recent references on the matter. The issue has been "hot" in the last few years and deserves more attention in the paper, such as numbers/percentages in mortalith and morbidity (and other consequences), as well as patophysioly of the MUCH. These topics should be approached in 1-2 additional paragraphs.

A: We have now revised the Introduction section (lines 37-42) and added new references (6,17,19,20,24,27,28-30,33).

Q: 3- If MUCH is "a clinical disease" or "a face of clinical disease" (Hypertension), are there advantages "to treat" this aspect of Hypertension? Although parts of items 2 and 3 have been briefly described, these topics should be better clarified in the form of a discussion. 

We have reported in the Discussion section the following sentence (lines 290-292): “Only a quarter of patients with MUCH, who given their ambulatory BP and risk should increase treatment (though this strategy is still under investigation in an ongoing trial [42]), were not identified.” Given the cardiovascular risk of patients with MUCH, they should undergo an increase in antihypertensive therapy guided by ambulatory BP and not by clinic BP. However, this type of approach is still under study.

Reviewer 2 Report

Coccina and colleagues propose an interesting study addressing possible prediction of masked uncontrolled hypertension through information, which can be gathered in a regular clinic (like demographics, office blood pressure, …). They analyzed a numerous Italian population, which underwent ABPM, trying to identify factors and construct models predicting masked uncontrolled hypertension. Authors were able to construct models that, with a reasonable AUC of almost 0.8, were able to predict masked hypertension.

The study theme is scientifically interesting and clinically very relevant. The study design is appropriate to answer the study question. The study and analysis both appear to have been carefully performed. The conclusions are sound with the design and the results.

The manuscript is clearly written, globally in a good English. However, its appeal might be improved if Authors were able to shorten it by approximately 5-10% of its current length (a couple of specific shortening proposals are listed below). The literature is well-done.

The main question, the Authors should try to answer (in their revised Discussion), is why their model should be preferred over the two others they are citing, which seemed to perform slightly better (at least in statistical terms). Geographical and ethnic factors may be part of the answer. (See specific comments below.)

Following specific (minor) comments must be addressed.

Line 74: “BP ≥135 and/or ≥85 mmHg”

1)    you probably rather mean SBP ≥135 and/or DBP ≥85mmHg.

2)    Whenever you use an abbreviation, at least for the first time, you need to define it (including very common abbreviations like BP, SBP, DBP, ...).

Line 121 (“Single, double and triple did not differ between the groups.”): What do you mean? Do you mean, first, second and third office readings?

Lines 124-128: “Results of univariate logistic regression analysis are shown in Table 3. Male sex, smoking habit, LV hypertrophy, clinic systolic BP in the range of 130-139 mmHg and clinic diastolic BP in the range of 85-89 mmHg were significantly associated with MUCH defined by both thresholds.” -> “In univariate logistic regression analysis, male sex, smoking, LV hypertrophy, clinic systolic BP in the range of 130-139 mmHg and clinic diastolic BP in the range of 85-89 mmHg were significantly associated with MUCH defined by both thresholds (Table 3).”

Line 130: “Results of multivariate logistic regression are reported in Table 4.”: delete.

Line 133: “associated with MUCH. The Hosmer” -> “associated with MUCH (Table 4). The Hosmer”

Lines 160 (“When the cutoff…”) up to Line 166 (“…were excluded.”): You already presented sens/spec (as well as AUC) above. Delete. (This will help you in shortening the manuscript.)

Line 167: “CI were 0.77, 0.06 and” -> “CI were 0.77±0.06 and”

Lines 180-181: “resulted accurate in identifying” -> “were appropriate in identifying”

Lines 184-193: Their score looks (slightly) better than yours. 1) Hypothesis of explanation? 2) What is the novelty and the interest of yours? Applicable to another population/geographical region (both cited models have been developed in Asian populations, while yours might be better in a Caucasian population)? Why not first of all explore the performance of Kim's score in your population, before developing an "ad hoc" (internally-developed in your population) prediction model? Even if not, it might be interesting to do an external validation of Kim's model in your population, and then compare the performance of your model and of Kim's model in your population. Your model might perform better in a Caucasian population. If you were able to prove it (even in a small sample), it would give a specific “flair” to your model and further increase the impact of your study and manuscript.

Lines 193-207: the same comments apply to the model developed and validated by Hung et al.

Lines 210-211 (“of all the studies”): either "of all the three studies" or "of all studies"

Lines 219-220: “risk should implement treatment” -> “risk be started on treatment”

Lines 222-223 (“we emphasize that our approach would allow to identify three-quarters of patients at higher risk rather than to not identify a quarter of them”):

1) reformulate in better English, e.g.: “we would like to stress that the proposed approach would at least allow to identify roughly three quarters of patients at risk, which represents already a big improvement over simply relying on resting office blood pressure”.

2) why should other countries use your model and not the ones of Kim or Hunt? For example (see also my comment above), your model might be the best suited in a Caucasian population, but the other two models might be preferable in Asian populations.

Line 227: “cannot be applied to other” -> “cannot be directly extrapolated to other”

Line 230: “or diabetes [34]. Third” -> “or diabetes mellitus [34]. Third”

Line 236: Please now list the strengths of your study (ideally, at least n=3). For example:

-       Pretty big sample size

-       Consistent findings (when comparing the 2 models and when comparing them with previous, similar studies)

-       Able to deliver a relatively easy-to-use model, which could be integrated in routine clinical practice.

-      

Line 238: “Male gender” -> “Male sex” (you are here interested in biological sex, not in gender identity)

Author Response

We thank the Reviewer very much for the kind and constructive revision.

Q: Its appeal might be improved if Authors were able to shorten it by approximately 5-10% of its current length.

A: We have tried to shorten the manuscript.

Q: The main question, the Authors should try to answer (in their revised Discussion), is why their model should be preferred over the two others they are citing, which seemed to perform slightly better (at least in statistical terms). Geographical and ethnic factors may be part of the answer.

A: We have now reported in the Discussion section (lines 283-286): “When we applied to our population the prediction models suggested by Kim et al.[34] and Hung et al.[35], their diagnostic accuracy tended to be lower. This aspect suggests that our model might be the best suited in a Caucasian population, but the other two models might be preferable in Asian populations. 

Q: Line 74: “BP ≥135 and/or ≥85 mmHg”

1)    you probably rather mean SBP ≥135 and/or DBP ≥85mmHg.

2)    Whenever you use an abbreviation, at least for the first time, you need to define it (including very common abbreviations like BP, SBP, DBP, ...).

A: We have now made the suggested changes (lines 69, 77-79).

Q: Line 121 (“Single, double and triple did not differ between the groups.”): What do you mean? Do you mean, first, second and third office readings?

A: We have now corrected (line 127): “Single, double and triple therapy…”.

Q: Lines 124-128: “Results of univariate logistic regression analysis are shown in Table 3. Male sex, smoking habit, LV hypertrophy, clinic systolic BP in the range of 130-139 mmHg and clinic diastolic BP in the range of 85-89 mmHg were significantly associated with MUCH defined by both thresholds.” -> “In univariate logistic regression analysis, male sex, smoking, LV hypertrophy, clinic systolic BP in the range of 130-139 mmHg and clinic diastolic BP in the range of 85-89 mmHg were significantly associated with MUCH defined by both thresholds (Table 3).”

A: We have now modified the sentence as suggested (lines 155-158).

Q: Line 130: “Results of multivariate logistic regression are reported in Table 4.”: delete.

A: We have now deleted the sentence.

Q: Line 133: “associated with MUCH. The Hosmer” -> “associated with MUCH (Table 4). The Hosmer”

A: We have now added “(Table 4)” (line 177).

Q: Lines 160 (“When the cutoff…”) up to Line 166 (“…were excluded.”): You already presented sens/spec (as well as AUC) above. Delete. (This will help you in shortening the manuscript.)

A: We have now deleted these sentences.

Q: Line 167: “CI were 0.77, 0.06 and” -> “CI were 0.77±0.06 and”

A: We have now made the suggested changes (lines 230 and 232).

Q: Lines 180-181: “resulted accurate in identifying” -> “were appropriate in identifying”

A: We have now changed the sentence (line 248).

Q: Lines 184-193: Their score looks (slightly) better than yours. 1) Hypothesis of explanation? 2) What is the novelty and the interest of yours? Applicable to another population/geographical region (both cited models have been developed in Asian populations, while yours might be better in a Caucasian population)? Why not first of all explore the performance of Kim's score in your population, before developing an "ad hoc" (internally-developed in your population) prediction model? Even if not, it might be interesting to do an external validation of Kim's model in your population, and then compare the performance of your model and of Kim's model in your population. Your model might perform better in a Caucasian population. If you were able to prove it (even in a small sample), it would give a specific “flair” to your model and further increase the impact of your study and manuscript.

A: We have now reported the following considerations in the final part of the Results (lines 238-239):            “ When we applied to our population the prediction model suggested by Kim et al.[34], the AUC for MUCH defined by 24-hour BP threshold was 0.73, 95% CI 0.69-0.77.”

We have also included an updated Table 1.

Q: Lines 193-207: the same comments apply to the model developed and validated by Hung et al.

A: We have now reported the following considerations in the final part of the Results (lines 239-243):               “If we used in our population the best prediction model suggested by Hung et al.[35], although they used a different definition of MUCH and also included patients with masked hypertension, the AUC for MUCH defined by 24-hour BP threshold (our definition that most closely matched their definition of MUCH) was 0.72, 95% CI 0.68-0.75.”

Q: Lines 210-211 (“of all the studies”): either "of all the three studies" or "of all studies"

A: We have now made the suggested change (line 277).  

Q: Lines 219-220: “risk should implement treatment” -> “risk be started on treatment”

A: We would like to report the following sentence (lines 290-292): “Only a quarter of patients with MUCH, who given their ambulatory BP and risk should increase treatment (though this strategy is still under investigation in an ongoing trial [42]), were not identified.”

Q: Lines 222-223 (“we emphasize that our approach would allow to identify three-quarters of patients at higher risk rather than to not identify a quarter of them”):

1) reformulate in better English, e.g.: “we would like to stress that the proposed approach would at least allow to identify roughly three quarters of patients at risk, which represents already a big improvement over simply relying on resting office blood pressure”.

A: We have now changed the sentence as suggested (lines 293-296).

Q: 2) why should other countries use your model and not the ones of Kim or Hunt? For example (see also my comment above), your model might be the best suited in a Caucasian population, but the other two models might be preferable in Asian populations.

A: We have now reported in the Discussion section (lines 283-286): “When we applied to our population the prediction models suggested by Kim et al.[34] and Hung et al.[35], their diagnostic accuracy tended to be lower. This aspect suggests that our model might be the best suited in a Caucasian population, but the other two models might be preferable in Asian populations. 

Q: Line 227: “cannot be applied to other” -> “cannot be directly extrapolated to other”

A: We have now changed the sentence as suggested (line 301).

Q: Line 230: “or diabetes [34]. Third” -> “or diabetes mellitus [34]. Third”

A: We have now added mellitus (line 304).

Q: Line 236: Please now list the strengths of your study (ideally, at least n=3). For example:

- Pretty big sample size

- Consistent findings (when comparing the 2 models and when comparing them with previous, similar studies)

- Able to deliver a relatively easy-to-use model, which could be integrated in routine clinical practice

A: We have now added the following sentence (lines 310-313): “Our study also has some strengths including 1) a fairly large sample size, 2) consistent findings when comparing the 2 models and when comparing them with similar studies, and 3) the ability to deliver a relatively easy-to-use model, which could be integrated into routine clinical practice.

Q: Line 238: “Male gender” -> “Male sex” (you are here interested in biological sex, not in gender identity)

A: We have now changed the term as suggested (line 315).